# Genetic control of *CCL24*, *POR*, and *IL23R* contributes to the pathogenesis of sarcoidosis

Akira Meguro et al.[#]

Sarcoidosis is a genetically complex systemic inflammatory disease that affects multiple organs. We present a GWAS of a Japanese cohort (700 sarcoidosis cases and 886 controls) with replication in independent samples from Japan (931 cases and 1,042 controls) and the Czech Republic (265 cases and 264 controls). We identified three loci outside the *HLA* complex, *CCL24*, *STYXL1-SRRM3*, and *C1orf141-IL23R*, which showed genome-wide significant associations ($P < 5.0 \times 10^{-8}$) with sarcoidosis; *CCL24* and *STYXL1-SRRM3* were novel. The disease-risk alleles in *CCL24* and *IL23R* were associated with reduced *CCL24* and *IL23R* expression, respectively. The disease-risk allele in *STYXL1-SRRM3* was associated with elevated *POR* expression. These results suggest that genetic control of *CCL24*, *POR*, and *IL23R* expression contribute to the pathogenesis of sarcoidosis. We speculate that the *CCL24* risk allele might be involved in a polarized Th1 response in sarcoidosis, and that *POR* and *IL23R* risk alleles may lead to diminished host defense against sarcoidosis pathogens.

---

[#]A list of authors and their affiliations appears at the end of the paper.

Sarcoidosis is a chronic multi-organ disease characterized by the presence of granulomas composed of epithelioid cells and activated T lymphocytes in multiple organs, including the lungs, skin, eyes, lymph nodes, liver, spleen, central and peripheral nervous systems, heart, salivary glands, and muscles[1]. Sarcoidosis occurs worldwide, but its prevalence and clinical features vary among different ethnic groups[1–3]. Although the etiology of sarcoidosis remains uncertain, its pathogenesis is thought to involve genetic factors[1].

Sarcoidosis is associated with human leukocyte antigen (HLA) class II genes, particularly HLA-DRB1[3–6]. BTNL2 variants in the HLA class II region have also been associated with sarcoidosis[4,6,7]. Genome-wide association studies (GWASs) in populations of European descent and/or African-Americans have reported several genetic loci/genes that confer susceptibility to sarcoidosis, including C10orf67, ANXA11, RAB23, OS9, CCDC88B, NOTCH4, and XAF1[8,9]. Recently, an Immunochip study identified four other loci associated with susceptibility to sarcoidosis in populations of European descent, but not African-Americans: SH2B3-ATXN2, IL12B, NFKB1-MANBA, and FAM117B[10]. Those findings could lead to a more precise understanding of the etiology and pathophysiology of sarcoidosis at the molecular level. However, no GWAS has yet been carried out in an Asian population.

Here, we describe the first GWAS of sarcoidosis performed in a Japanese population. We aimed to confirm and identify susceptibility genes, particularly outside the HLA complex. Our analysis identifies three non-HLA susceptibility loci (CCL24, STYXL1-SRRM3, and C1orf141-IL23R) for sarcoidosis and demonstrates the importance of genetic control of CCL24, POR, and IL23R, through genetic polymorphisms, in the pathogenesis of sarcoidosis.

## Results

### GWAS discovery stage
We analyzed 530,466 autosomal SNPs from 685 patients and 847 controls after performing sample and SNP quality control procedures (see "Genome-wide association study and quality control procedures" section). The genomic inflation factor ($\lambda$) values were 1.060 and 1.056, with and without HLA region SNPs, respectively (Supplementary Fig. 1). This result indicated that population stratification had a negligible effect. A principal component analysis showed that the cases and controls in the GWAS discovery stage were genetically well matched (Supplementary Fig. 2).

The HLA class II region displayed the most significant association with sarcoidosis (rs9274741; the P-value was corrected for the first five ancestry principal components and genomic inflation; $[P_{GC}] = 2.5 \times 10^{-16}$, odds ratio [OR] = 2.34, 95% confidence interval [CI] = 1.90–2.88). We identified 17 SNPs in the HLA complex that reached the genome-wide significance threshold in or near the following four genes: NOTCH4, BTNL2, HLA-DQB1, and HLA-DQA2 (Fig. 1; Supplementary Table 1). A stepwise regression analysis showed that variants of these four gene regions were independently associated with sarcoidosis (Supplementary Table 1). Of the HLA alleles, only HLA-DRB1*0803 showed genome-wide significance as a risk factor for sarcoidosis ($P_{GC} = 4.1 \times 10^{-8}$, OR = 1.82, 95% CI = 1.46–2.26); HLA-DQB1*0601 showed the next highest significance as a risk allele for sarcoidosis ($P_{GC} = 1.5 \times 10^{-7}$, OR = 1.32, 95% CI = 1.12–1.56). In contrast, HLA-B*0702, HLA-DRB1*0101, and HLA-DQB1*0501 exhibited genome-wide significance as protective alleles (Supplementary Table 2). These associations between HLA class II alleles and sarcoidosis were consistent with previous findings in Japanese populations[3,4]. Indeed, the association between HLA-B*0702 and sarcoidosis was first identified in a Japanese population; in contrast, HLA-B*07 was associated with increased risk of sarcoidosis in a Swedish population[5]. Significant SNPs in the 6p21.32–6p21.33 region that included these genes were not analyzed in our replication study, because it had been well-established that sarcoidosis was strongly associated with that genomic region.

Importantly, outside the HLA complex, an additive genetic model indicated that 2019 distinct loci were associated with sarcoidosis at $P_{GC} < 0.01$. However, no SNPs cleared the genome-wide significance threshold ($P_{GC} < 5.0 \times 10^{-8}$). The strongest association was observed in LOC102724036 on 9q21.33 (rs7871278 $P_{GC} = 1.6 \times 10^{-7}$, OR = 1.52, 95% CI = 1.31–1.76; Fig. 1). The second strongest association was observed in the 400 kb region extending from CCL24 to SRRM3 on 7q11.23 (rs11983987 $P_{GC} = 2.7 \times 10^{-7}$, OR = 1.50, 95% CI = 1.29–1.74; Fig. 1). Two loci (ANXA11 and CCDC88B) that were associated with sarcoidosis in previous GWASs showed a $P_{GC} < 0.01$ in our Japanese discovery set (lead SNP: ANXA11 rs1049550: $P_{GC} = 0.0036$, OR = 1.24, 95% CI = 1.07–1.43; CCDC88B rs11231740: $P_{GC} = 0.0047$, OR = 1.27, 95% CI = 1.07–1.50; Supplementary Data 1). In addition, one locus from a previous Immunochip study (FAM117B) was marginally associated with sarcoidosis in our discovery set (lead SNP: rs10755019: $P_{GC} = 0.018$, OR = 1.25, 95% CI = 1.05–1.48; Supplementary Data 1).

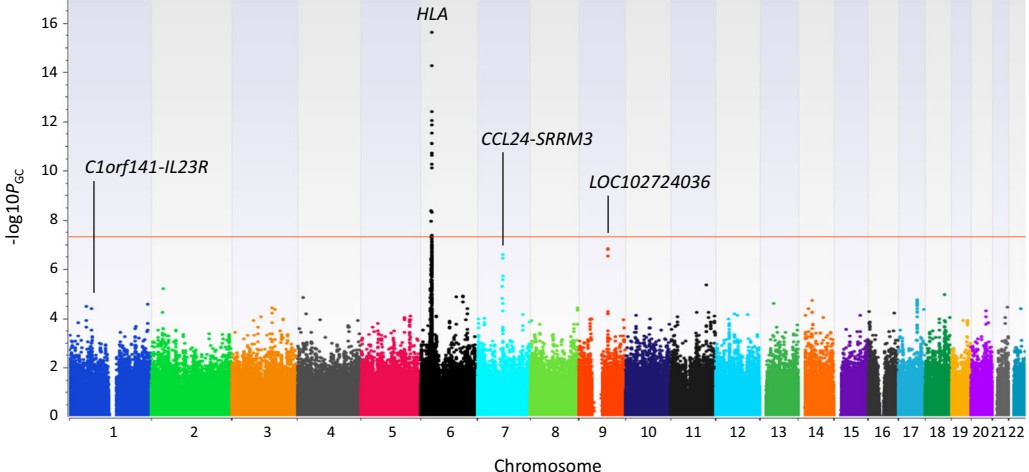

**Fig. 1 Genome-wide association results for 685 cases of sarcoidosis and 847 controls from the Japanese population.** The −log10 ($P_{GC}$) values for 530,466 autosomal SNPs are shown according to their corresponding chromosomes, and they are sorted by genomic position. Chromosomes are indicated with alternating colors. The horizontal red line indicates the genome-wide significance threshold of $P_{GC} = 5.0 \times 10^{-8}$.

**Meta-analysis of the GWAS discovery and replication cohorts**.
To validate the GWAS results, the lead SNPs from each of the 2,019 identified loci were tested in replication studies with independent Japanese and Czech samples. We identified three loci (*CCL24*, *STYXL1-SRRM3*, and *C1orf141-IL23R*) that showed a genome-wide significant association with sarcoidosis in the meta-analysis of the GWAS cohort and the two replication cohorts (lead genotyped SNP: *CCL24* rs2302006: $P_{GC} = 1.2 \times 10^{-8}$, OR = 1.31, 95% CI = 1.19–1.44; *STYXL1-SRRM3* rs3779419: $P_{GC} = 1.8 \times 10^{-10}$, OR = 1.37, 95% CI = 1.24–1.51; *C1orf141-IL23R* rs3762318: $P_{GC} = 3.1 \times 10^{-10}$, OR = 1.65, 95% CI = 1.41–1.92; Table 1). Of these three loci, *C1orf141-IL23R* had been previously reported as a sarcoidosis-related locus[10–12]; however, *CCL24* and *STYXL1-SRRM3* represented novel susceptibility loci for sarcoidosis. The most significant SNP outside the *HLA* complex observed in the GWAS discovery stage, *LOC102724036* rs7871278, was not replicated in either of the two replication cohorts (Japanese: $P = 0.23$, OR = 0.92, 95% CI = 0.81–1.05; Czech: $P = 0.29$, OR = 1.16, 95% CI = 0.88–1.52) ($P_{GC} = 0.0064$, OR = 1.14, 95% CI = 1.04–1.25 across all three cohorts).

We then imputed the *CCL24*, *STYXL1-SRRM3*, and *C1orf141-IL23R* loci and found imputed SNPs with stronger associations than the lead genotyped SNP in each locus (Table 1 and Fig. 2). In the *CCL24* locus, the intronic SNP, rs4728493, showed the most significant association with disease in the meta-analysis of the GWAS and replication cohorts ($P_{GC} = 1.1 \times 10^{-11}$, OR = 1.39, 95% CI = 1.27–1.53) with a significance level of $P < 0.05$ in all three cohorts (Table 1 and Fig. 2a). We observed moderate heterogeneity ($I^2 = 46.1\%$, $P = 0.16$) among the three cohorts. The *STYXL1-SRRM3* locus, which includes several genes (*POR*, *TMEM120A*, *STYXL1*, *MDH2*, *LOC101927126*, *SRRM3*, and *HSPB1*), is adjacent to the *CCL24* locus on chromosome 7 (Fig. 2b and Supplementary Fig. 3). In this locus, the *TMEM120A* intronic SNPs, rs112463197 and rs112680895, showed the most significant association with disease in the meta-analysis across all three cohorts ($P_{GC} = 1.3 \times 10^{-10}$, OR = 1.37, 95% CI = 1.25–1.51), with low heterogeneity ($I^2 = 18.0\%$, $P = 0.30$); although these variants did not attain significance in the Czech cohort, the direction of their allelic effect on disease risk was consistent between the Japanese (OR = 1.29–1.52) and Czech cohorts (OR = 1.26; Table 1 and Fig. 2b). The association peaks in the *STYXL1-SRRM3* locus were located ~200–400 kb from the *CCL24* locus; however, we found that the *STYXL1-SRRM3* lead SNPs, rs112463197 and rs112680895, showed no linkage disequilibrium (LD) with the *CCL24* lead SNP (rs4728493) (Japanese: $D' = 0.01$, $r^2 = 0.00$; Czech: $D' = 0.04$, $r^2 = 0.00$; Supplementary Fig. 3). Indeed, after conditioning on rs4728493, we found that rs112463197 and rs112680895 were associated with sarcoidosis, independent of *CCL24* variants ($P_{GC} = 8.7 \times 10^{-11}$ across all three cohorts).

In the *C1orf141-IL23R* locus, the SNPs, rs117633859 and rs117282985, located ~4 kb upstream of *IL23R*, showed the most significant association in the meta-analysis across all three cohorts ($P_{GC} = 2.0 \times 10^{-12}$, OR = 1.87, 95% CI = 1.57–2.23), with low heterogeneity ($I^2 = 16.8\%$, $P = 0.30$). These two variants were strongly associated with sarcoidosis in the two Japanese cohorts ($P < 1.0 \times 10^{-5}$). Although they did not attain significance in the Czech cohort, the direction of the effect of these risk alleles was consistent between the Japanese (OR = 1.88–2.01) and Czech cohorts (OR = 1.21; Table 1 and Fig. 2c). The *IL23R* rs11209026 (Arg381Glu) was previously reported to be associated with chronic sarcoidosis in a European cohort[11]. Another study with mixed ethnicities (including mainly populations of European descent) reported that *IL23R* rs11209026 and rs11465804 were associated with sarcoid uveitis, and that *IL23R* rs7517847 was associated with sarcoidosis without uveitis[12]. In the current study,

these three SNPs were not significantly associated with sarcoidosis in our Japanese or Czech cohorts; also, rs11465804 and rs11209026 were monomorphic in the Japanese cohort (Supplementary Table 3). Furthermore, these three SNPs were not in LD with rs117633859 in either population (Japanese: $D' \leq 0.33$, $r^2 \leq 0.01$; Czech: $D' \leq 0.37$, $r^2 \leq 0.03$; Supplementary Fig. 4). The recent Immunochip study showed that rs12069782 in *C1orf141* was strongly associated with sarcoidosis in German and Swedish populations, but not in Czech or African-American populations[10]. Our study found that rs12069782 had the second strongest association with disease in the locus of our Japanese cohorts ($P_{GC} = 1.1 \times 10^{-11}$, OR = 1.89, 95% CI = 1.56–2.25 across two Japanese cohorts), but this SNP was not significantly associated with disease in our Czech cohort (Supplementary Table 3), which included most of the Czech individuals used in the Immunochip study. Moreover, rs12069782 was in strong LD with rs117633859 in the Japanese cohort, but not in the Czech cohort (Japanese: $D' = 0.97$, $r^2 = 0.93$; Czech: $D' = 1.00$, $r^2 = 0.10$; Supplementary Fig. 4). In our Czech cohort, the intronic SNPs, rs6664119 and rs11209013, had the lowest *P*-values in the locus for association with disease ($P = 0.0048$, OR = 1.42, 95% CI = 1.11–1.82) ($P_{GC} = 4.5 \times 10^{-7}$, OR = 1.26, 95% CI = 1.15–1.38 across all three cohorts; Table 1). rs6664119 and rs11209013 were more strongly linked to rs3762318 and rs12069782 than with any other SNPs, including rs117633859, in the Czech cohort (Supplementary Fig. 4).

To identify the most plausible SNPs within the identified loci, we performed a fine-mapping analysis with FINEMAP[13], based on the results from the GWAS discovery cohort (Fig. 2; Supplementary Data 2). Fine-mapping of *CCL24* indicated that the rs4728493 had the highest posterior inclusion probability (0.1179) of being the causal variant at the locus. At *STYXL1-SRRM3*, rs112463197 and rs112680895 had the highest posterior inclusion probability (0.0649). At *C1orf141-IL23R*, rs117633859 and rs117282985 had the highest posterior inclusion probability (0.0765), which was nearly threefold larger than the second-highest posterior inclusion probability (0.0243 for 17129680).

**eQTL analysis of *CCL24*, *STYXL1-SRRM3*, and *IL23R***. The disease-associated SNPs in the *CCL24*, *STYXL1-SRRM3*, and *IL23R* loci were located in non-coding regions. Because non-coding variants can significantly affect the expression of genes, we performed an eQTL analysis of *CCL24*, *STYXL1-SRRM3*, and *IL23R* (Supplementary Table 4). Our eQTL analysis revealed that the disease-risk alleles in *CCL24* (the C allele of rs4728493) and *IL23R* (the G allele of rs117633859 and the C allele of rs6664119) were significantly associated with reduced expression of *CCL24* and *IL23R*, respectively, in whole blood. Publicly available data from the GTEx Portal[14] showed a significant association between reduced *CCL24* expression and the risk allele of *CCL24* in cultured fibroblasts, ovary, tibial nerves, and skin ($P_{\text{multi-tissue meta-analysis}} = 1.1 \times 10^{-19}$ in a meta-analysis of tissues/organs tested in the GTEx Portal; Supplementary Fig. 5). For *STYXL1-SRRM3*, both our eQTL analysis and the GTEx Portal showed that the disease-risk allele (the T allele of rs112463197) was significantly associated with increased *POR* expression. Moreover, the GTEx Portal demonstrated that the T allele had an eQTL effect on *POR* in many tissues/organs ($P_{\text{multi-tissue meta-analysis}} = 5.6 \times 10^{-184}$; Supplementary Fig. 6).

We then performed a colocalization analysis with eCAVIAR[15] to identify potentially causal SNPs that were responsible for both the GWAS and eQTL signals (Supplementary Table 5). The *CCL24* locus had four SNPs (rs4728493, rs62477637, rs62477640, and rs7802368) that showed significant colocalizations, with a colocalization posterior probability (CLPP) > 0.01. The GWAS

**Table 1 Associations with sarcoidosis found in the *CCL24*, *STYXL1-SRRM3*, and *C1orf141-IL23R* loci across all three cohorts.**

| SNP | Chr. | Position (Build 37.1) | Nearest gene | Risk allele | Population | N | | Risk allele frequency | | P | $P_{GC}$ | OR (95% CI) | Heterogeneity | |
|---|---|---|---|---|---|---|---|---|---|---|---|---|---|---|
| | | | | | | Cases | Controls | Cases | Controls | | | | $I^2$, % | P |
| rs2302006 | 7 | 75,442,730 | CCL24 | A | GWAS, Japanese | 685 | 847 | 0.525 | 0.443 | 5.9E−06 | 1.6E−05 | 1.40 (1.21–1.63) | | |
| | | | | | Replication, Japanese | 907 | 1,042 | 0.530 | 0.479 | 0.0015 | | 1.23 (1.08–1.39) | | |
| | | | | | Replication, Czech | 252 | 256 | 0.849 | 0.793 | 0.018 | | 1.48 (1.07–2.06) | | |
| | | | | | Meta-analysis | | | | | 7.2E−09 | 1.2E−08 | 1.31 (1.19–1.44) | 18.0 | 0.30 |
| rs4728493 | 7 | 75,446,974 | CCL24 | C | GWAS, Japanese | 685 | 847 | 0.675 | 0.580 | 2.4E−07 | 1.9E−06 | 1.49 (1.28–1.73) | | |
| | | | | | Replication, Japanese | 907 | 1,042 | 0.638 | 0.570 | 2.9E−05 | | 1.31 (1.15–1.48) | | |
| | | | | | Replication, Czech | 252 | 256 | 0.942 | 0.896 | 0.0070 | | 1.91 (1.19–3.07) | | |
| | | | | | Meta-analysis | | | | | 5.7E−12 | 1.1E−11 | 1.39 (1.27–1.53) | 46.1 | 0.16 |
| rs112463197[a] | 7 | 75,622,912 | TMEM120A | T | GWAS, Japanese | 685 | 847 | 0.390 | 0.299 | 5.3E−08 | 1.1E−07 | 1.52 (1.31–1.78) | | |
| | | | | | Replication, Japanese | 907 | 1,042 | 0.384 | 0.325 | 1.3E−04 | | 1.29 (1.13–1.48) | | |
| | | | | | Replication, Czech | 252 | 256 | 0.185 | 0.152 | 0.17 | | 1.26 (0.90–1.75) | | |
| | | | | | Meta-analysis | | | | | 6.3E−11 | 1.3E−10 | 1.37 (1.25–1.51) | 18.0 | 0.30 |
| rs3779419 | 7 | 75,695,081 | MDH2 | A | GWAS, Japanese | 685 | 847 | 0.389 | 0.300 | 2.6E−07 | 3.9E−07 | 1.49 (1.28–1.73) | | |
| | | | | | Replication, Japanese | 907 | 1,042 | 0.386 | 0.324 | 6.5E−05 | | 1.31 (1.15–1.49) | | |
| | | | | | Replication, Czech | 252 | 256 | 0.188 | 0.156 | 0.17 | | 1.26 (0.90–1.74) | | |
| | | | | | Meta-analysis | | | | | 9.0E−11 | 1.8E−10 | 1.37 (1.24–1.51) | 0.0 | 0.42 |
| rs3762318 | 1 | 67,597,119 | C1orf141 | C | GWAS, Japanese | 685 | 847 | 0.107 | 0.064 | 2.7E−05 | 4.1E−05 | 1.75 (1.34–2.27) | | |
| | | | | | Replication, Japanese | 907 | 1,042 | 0.103 | 0.054 | 2.7E−08 | | 1.97 (1.55–2.52) | | |
| | | | | | Replication, Czech | 252 | 256 | 0.222 | 0.197 | 0.33 | | 1.16 (0.86–1.57) | | |
| | | | | | Meta-analysis | | | | | 2.0E−10 | 3.1E−10 | 1.65 (1.41–1.92) | 73.1 | 0.024 |
| rs117633859[a] | 1 | 67,627,828 | IL23R | G | GWAS, Japanese | 685 | 847 | 0.109 | 0.059 | 2.8E−06 | 6.5E−06 | 1.88 (1.44–2.46) | | |
| | | | | | Replication, Japanese | 907 | 1,042 | 0.103 | 0.053 | 1.2E−08 | | 2.01 (1.57–2.56) | | |
| | | | | | Replication, Czech | 252 | 256 | 0.048 | 0.040 | 0.54 | | 1.21 (0.66–2.22) | | |
| | | | | | Meta-analysis | | | | | 1.1E−12 | 2.0E−12 | 1.87 (1.57–2.23) | 16.8 | 0.30 |
| rs6664119[a] | 1 | 67,655,895 | IL23R | C | GWAS, Japanese | 685 | 847 | 0.443 | 0.394 | 0.013 | 0.021 | 1.20 (1.04–1.39) | | |
| | | | | | Replication, Japanese | 907 | 1,042 | 0.428 | 0.370 | 3.1E−04 | | 1.26 (1.11–1.43) | | |
| | | | | | Replication, Czech | 252 | 256 | 0.520 | 0.430 | 0.0048 | | 1.42 (1.11–1.82) | | |
| | | | | | Meta-analysis | | | | | 3.9E−07 | 4.5E−07 | 1.26 (1.15–1.38) | 0.0 | 0.57 |

[a]rs112680895, rs117282985, and rs11209013 had the same association results as rs112463197, rs117633859, and rs6664119, respectively.

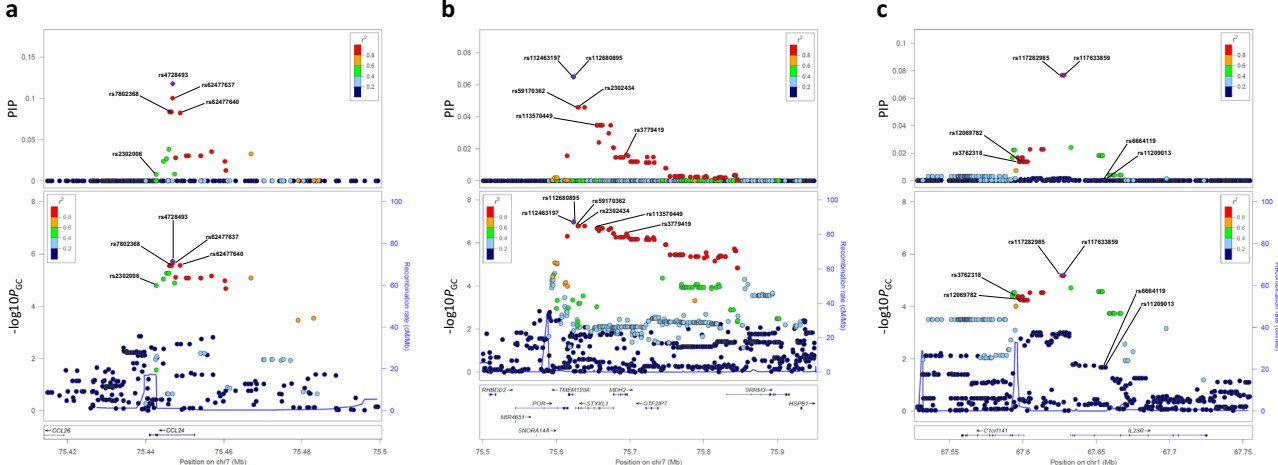

**Fig. 2 In-depth SNP analysis of the three candidate gene regions in the Japanese GWAS discovery cohort.** Data are shown for **a** *CCL24*, **b** *STYXL1-SRRM3*, and **c** *C1orf141-IL23R*. (Top row) Posterior inclusion probability (PIP) for each SNP, obtained with fine-mapping. (Middle row) Regional association plots for each region. The left *y* axes represent the −log10 ($P_{GC}$) values for associations with sarcoidosis; the right *y* axes (blue lines) represent the estimated recombination rate. The lead SNP in each region is depicted as a purple diamond. The color coding for all other SNPs indicates linkage disequilibrium with the lead SNP, as follows: red, $r^2 \geq 0.8$; yellow, $0.6 \leq r^2 < 0.8$; green, $0.4 \leq r^2 < 0.6$; cyan, $0.2 \leq r^2 < 0.4$; blue, $r^2 < 0.2$; and gray, $r^2$ unknown. (Bottom row) Gene annotations.

lead SNP, rs4728493, had the highest CLPP score (0.0199) among the tested SNPs. At the *STYXL1-SRRM3* locus, we also found four SNPs (rs59170362, rs2302434, rs113570449, and rs112463197) that showed significant colocalizations. Three of these SNPs (rs59170362, rs2302434, and rs113570449) had higher CLPPs (0.112–0.0125) than the GWAS lead SNP, rs112463197 (CLPP = 0.0103). The SNPs in the *CCL24* and *STYXL1-SRRM3* loci that were newly identified in the colocalization analysis were in strong LD with the GWAS lead SNPs in each locus (*CCL24*: $D' > 0.99$, $r^2 > 0.99$; *STYXL1-SRRM3*: $D' > 0.99$, $r^2 > 0.98$ in the Japanese cohort). Moreover, these SNPs also showed genome-wide significant associations ($P_{GC} < 5.0 \times 10^{-8}$) with sarcoidosis in the meta-analysis. However, the GWAS lead SNPs showed the lowest $P_{GC}$-values (i.e., the strongest association with sarcoidosis) in each of these loci. At the *IL23R* locus, only the GWAS lead SNP rs117633859 showed significant colocalization (CLPP = 0.0109).

**Stratified analysis by CXR stage and Löfgren's syndrome.** Table 2 shows the results of the association analysis of the disease-risk alleles in *CCL24*, *STYXL1-SRRM3*, and *IL23R* after stratifying according to the chest X-ray (CXR) stage and the presence of Löfgren's syndrome. The OR of the *CCL24* rs4728493 C allele significantly declined according to an incline in CXR stages: 0–I, II, and III–IV, in both the Japanese and Czech cohorts (Mantel-extension test: $P = 0.045$). In the Czech cohort, the C allele had the highest OR (3.11, 95% CI = 0.94–10.32) in patients with Löfgren's syndrome. We also found that the *STYXL1-SRRM3* rs112463197 T allele had the highest OR (2.08, 95% CI = 1.19–3.64) in patients with Löfgren's syndrome in the Czech cohort, and it had a higher OR in patients with CXR stages 0–II than in patients with stages III–IV in both the Japanese and Czech cohorts, although the trend was not significant. On the other hand, the ORs of the risk alleles in *IL23R* were not affected by the presence of Löfgren's syndrome or CXR stages.

**Two loci from previous GWASs (*ANXA11* and *CCDC88B*).** *ANXA11* rs1049550 was associated with sarcoidosis in both the Japanese and Czech replication cohorts ($P = 0.0098$ and $P = 1.8 \times 10^{-4}$, respectively); however, it did not clear the genome-wide

significance threshold in the meta-analysis of the GWAS cohort and the two replication cohorts ($P_{GC} = 1.5 \times 10^{-6}$, OR = 1.25, 95% CI = 1.14–1.37, with high heterogeneity: $I^2 = 58.7\%$, $P = 0.089$, across all three cohorts; Supplementary Table 6). *CCDC88B* rs11231740 was associated with sarcoidosis only in the Czech cohort ($P = 0.0058$, OR = 1.42, 95% CI = 1.11–1.82). We found no allelic effect of *CCDC88B* rs11231740 on disease risk in the Japanese replication cohort ($P_{GC} = 8.8 \times 10^{-4}$, OR = 1.19, 95% CI = 1.08–1.33, with high heterogeneity: $I^2 = 54.7\%$, $P = 0.11$, across all three cohorts; Supplementary Table 6).

**Discussion**

To the best of our knowledge, this was the first GWAS in sarcoidosis performed in an East-Asian, Japanese population. We identified three loci, *CCL24*, *STYXL1-SRRM3*, and *C1orf141-IL23R*, which showed genome-wide significant associations ($P_{GC} < 5.0 \times 10^{-8}$) with the disease in a meta-analysis of the GWAS and replication cohorts, which included a European, Czech population. Importantly, two of the three identified loci, *CCL24* and *STYXL1-SRRM3*, were novel in the context of sarcoidosis. Moreover, all three loci showed functional relevance; the *CCL24* and *IL23R* risk variants were associated with reduced *CCL24* and *IL23R* expression, respectively; and the *STYXL1-SRRM3* risk variant was associated with elevated *POR* expression.

By interacting with CCR3, CCL24 (C-C motif chemokine ligand 24, also known as eotaxin-2) is an important mediator in Th2 cell-mediated allergic inflammation occurring in asthma, allergic rhinitis, and atopic dermatitis. However, a highly polarized Th1 cytokine profile at disease sites is a typical immunological feature of sarcoidosis[16]. Indeed, Th1 cytokines, such as IFN-γ, TNF-α, IL-2, IL-12, and IL-18, have important roles in sarcoid granuloma formation[1,17]. The current study was the first to describe an association between *CCL24* variants and sarcoidosis. We identified an intronic SNP, rs4728493, in the *CCL24* region. The risk allele (C) of rs4728493 was significantly associated with reduced *CCL24* expression. Reduced *CCL24* expression would downregulate the Th2 response; thus, our findings suggested that the C allele may contribute to a Th1-biased immune response.

The Th1/Th2 balance has been shown to shift, during the course of sarcoidosis, toward an immune response characterized by Th2 cytokines[1,16]. This shift suggested that the transition from

**Table 2 Associations with sarcoidosis found for CCL24 rs4728493, STYXL1-SRRM3 rs112463197, and IL23R rs117633859/rs6664119 in populations stratified according to chest X-ray (CXR) stage and Löfgren's syndrome.**

| Population | Status | N | CCL24 rs4728493 | | | STYXL1-SRRM3 rs112463197 | | | IL23R rs117633859/rs6664119[b] | | |
|---|---|---|---|---|---|---|---|---|---|---|---|
| | | | Risk allele (C) freq. | P[a] | OR (95% CI) | Risk allele (T) freq. | P[a] | OR (95% CI) | Risk allele freq. | P[a] | OR (95% CI) |
| Japanese | Controls | 1,889 | 57.5 | | | 31.2 | | | 5.7 | | |
| | Cases[c] | 640 | 63.5 | 2.0E-04 | 1.29 (1.13-1.48) | 39.8 | 5.5E-08 | 1.46 (1.27-1.66) | 11.0 | 2.8E-10 | 2.08 (1.66-2.61) |
| | CXR stage 0 + I | 334 | 65.4 | 1.6E-04 | 1.41 (1.18-1.68) | 40.7 | 2.7E-06 | 1.52 (1.28-1.80) | 11.1 | 4.0E-07 | 2.06 (1.56-2.73) |
| | Stage II | 219 | 61.9 | 0.084 | 1.20 (0.98-1.48) | 40.0 | 3.0E-04 | 1.47 (1.20-1.81) | 11.0 | 2.3E-05 | 2.06 (1.48-2.88) |
| | Stage III + IV | 87 | 60.3 | 0.46 | 1.13 (0.83-1.54) | 35.6 | 0.23 | 1.22 (0.89-1.68) | 10.9 | 0.0051 | 2.04 (1.24-3.36) |
| Czech | Controls | 256 | 89.6 | | | 15.2 | | | 43.0 | | |
| | Cases | 247 | 94.1 | 0.0082 | 1.90 (1.17-3.07) | 18.2 | 0.21 | 1.24 (0.89-1.72) | 52.0 | 0.0048 | 1.42 (1.11-1.82) |
| | CXR stage 0 + I | 115 | 94.8 | 0.020 | 2.14 (1.11-4.14) | 17.8 | 0.37 | 1.21 (0.80-1.85) | 47.8 | 0.23 | 1.21 (0.89-1.64) |
| | Stage II | 113 | 93.8 | 0.066 | 1.78 (0.96-3.32) | 19.0 | 0.21 | 1.30 (0.87-1.95) | 55.3 | 0.0023 | 1.63 (1.19-2.24) |
| | Stage III + IV | 19 | 92.1 | 0.62 | 1.36 (0.40-4.65) | 15.8 | 0.93 | 1.04 (0.42-2.59) | 52.6 | 0.25 | 1.47 (0.76-2.84) |
| | Löfgren's syndrome | 41 | 96.3 | 0.052 | 3.11 (0.94-10.32) | 26.8 | 0.0090 | 2.08 (1.19-3.64) | 52.4 | 0.11 | 1.46 (0.91-2.33) |

[a]$P_{GC}$ and P-values were used to test for significance in the Japanese and Czech cohorts, respectively.
[b]The G allele of rs117633859 and the C allele of rs6664119 are risk alleles for sarcoidosis in the Japanese and Czech populations, respectively.
[c]Patients that had sarcoidosis for more than 5 years were selected for this analysis to ensure an established CXR stage. No patient had Löfgren's syndrome in the Japanese population.

Th1 to Th2 might have a critical role in the development of the persistent form of the disease, which progresses to fibrosis. Bronchoalveolar fluid samples from patients with stage III disease (based on CXRs) showed increased CCL24 levels compared to samples from patients with Löfgren's syndrome, which is a subtype of acute sarcoidosis that has a better prognosis than other disease subtypes[18]. Recent studies suggested that the Th2 response and CCL24 were most likely involved in the development and persistence of the disease[19,20]. Our study found that the OR of the rs4728493 C allele declined according to an incline in CXR stages: 0–I, II, and III–IV, and that the C allele had the highest OR in patients with Löfgren's syndrome. These findings suggested that the C allele might be involved in a polarized Th1 response in the early stages of sarcoidosis and in the acute disease. Moreover, the C allele might contribute to suppressing disease (fibrosis) progression through its effect on CCL24 expression. This hypothesis was supported by the following findings: (a) CCL24 stimulated lung fibroblast proliferation[21]; (b) blockade of CCL24 inhibited skin and lung inflammation and fibrosis[22]; and (c) eQTL analysis demonstrated the reduced CCL24 expression associated with the C allele in cultured fibroblasts and skin.

POR (cytochrome p450 oxidoreductase) is a flavoprotein that transfers electrons from NADPH to all microsomal (type 2) cytochrome P450 enzymes, including steroidogenic P450c17, P450c21, and P450aro. Thus, POR has a direct role in steroidogenesis[23]. Mutations in POR cause diseases associated with abnormal steroidogenesis, such as congenital adrenal hyperplasia[24] and Antley–Bixler syndrome[25]; however, the relationship between POR and sarcoidosis has not yet been elucidated. Our study found that the risk T allele of a sarcoidosis-associated SNP, rs112463197, in the STYXL1-SRRM3 locus was associated with increased POR expression in many of the organs affected by sarcoidosis, including the lungs, skin, sigmoid colon, esophagus, minor salivary gland, kidney, spleen, heart, nerves, pituitary, spinal cord, and liver. This increase could lead to the enhancement of endogenous steroidogenesis, which was reported to induce immunosuppression and result in increased susceptibility to infectious agents[26,27]. Various pathogens have been suggested to contribute to sarcoidosis development; recent studies have focused on the role of mycobacterial or propionibacterial organisms in the disease etiology[1,16]. Therefore, the T allele of rs112463197 might induce immunosuppression, due to enhanced endogenous steroid levels, and lead to diminished host defense against pathogens involved in the development of sarcoidosis. In our study, the rs112463197 T allele showed high ORs in patients with Löfgren's syndrome and in patients with CXR stages 0–II. These patients generally have a good prognosis, or a high/moderate rate of spontaneous remission[2], compared to patients with stage III or IV, who exhibit little or no spontaneous remission[2]. The symptoms of sarcoidosis often ameliorate during pregnancy, due to the favorable effect of increased levels of endogenous steroid hormones, such as corticosteroids[28–30]. Therefore, the T allele might contribute to spontaneous sarcoidosis remission, due to its effect on endogenous steroid levels, but it might also be a risk factor for disease development.

IL23R (interleukin 23 receptor) encodes a subunit of the receptor for IL-23. IL-23 has been shown to stimulate Th17 cell proliferation and increase the production of inflammatory cytokines (e.g., IL-1, IL-6, IL-17, and TNFα)[31]. Because Th17 cells induce neutrophil and macrophage chemotaxis, they have important roles in host defense against extracellular pathogens and in the development of immune-mediated diseases[32]. Recent studies have identified IL23R as a susceptibility gene for several inflammatory/immune-mediated diseases[33–37]. The IL23R SNPs identified in Crohn's disease, ulcerative colitis, psoriasis, ankylosing spondylitis, and Behçet's disease were located in different

LD blocks from the LD blocks of the sarcoidosis-associated SNP (rs117633859) identified in this study (Supplementary Fig. 4). Moreover, those *IL23R* SNPs were not associated with sarcoidosis in our Japanese or Czech cohorts (Supplementary Table 3). Only one SNP associated with Crohn's disease and ulcerative colitis, rs76418789 (*IL23R* Gly149Arg), was significantly associated with sarcoidosis (*P* < 0.05) in both our Japanese discovery and replication cohorts. However, the ORs indicated opposite associations (0.64 vs. 1.42) in the two Japanese cohorts (Supplementary Table 3). Interestingly, in our Japanese population, one of the SNPs we identified was previously associated with Vogt–Koyanagi–Harada (VKH) disease in Han Chinese patients[37]. This SNP, rs117633859, showed the strongest association with both sarcoidosis and VKH disease[37] of all the SNPs in the *IL23R* locus. Moreover, the G allele, which was associated with reduced *IL23R* expression, conferred risk for both diseases. Therefore, the rs117633859 G allele might downregulate *IL23R* expression, and carriers might have a diminished Th17 response, and thus, a diminished host defense against pathogens. This mechanism might explain the link between rs117633859 and sarcoidosis development. Indeed, stimulation with *Propionibacterium acnes* elicited significantly lower *IL17* expression in peripheral blood mononuclear cells from patients with sarcoidosis than in those from healthy controls. Conversely, the expression levels of Th1 cytokine genes, *IL2* and *IL12*, were significantly higher in patients than in controls[38]. Those findings suggested that sarcoidosis might arise from an imbalance in Th1/Th17 immune responses to sarcoidosis-associated pathogens.

All of the inflammatory/immune-mediated diseases mentioned above that were associated with *IL23R* variants are also associated with *HLA* genes[33–37]. Therefore, these diseases may share a common pathogenic pathway via *IL23R* and *HLA*. In particular, sarcoidosis and VKH are both associated with granulomatous uveitis. They share the same *IL23R* variant (rs117633859), and they are both associated with class II *HLA* polymorphisms (sarcoidosis: *HLA-DRB1*03*, **08*, **09*, **11*, **12*, **14*, and **15*; VKH disease: *HLA-DRB1*04*) as risk factors.

A limitation of our study was that only the identified loci were explored with imputation; thus, the whole genome data were not imputed. Therefore, we may have missed some potential loci that conferred susceptibility to sarcoidosis. In future studies, an imputation analysis based on our GWAS results is needed for a comprehensive clarification of the genetic factors associated with sarcoidosis. On the other hand, replication cohorts from two distinct populations and of sizes quite substantial in context of sarcoidosis represent strengths of our study.

Our study revealed substantial differences in the allele distribution of numerous SNPs between Asian Japanese and Caucasian Czech controls, even for the *CCL24* lead SNP associated with sarcoidosis (Table 1). This finding indirectly supported the concept that differences in genetic background can determine ethnic differences in sarcoidosis prevalence and phenotypes. To our knowledge, this study was the first GWAS of sarcoidosis in an Asian population. We found three genome-wide significant non-*HLA* loci, *CCL24*, *STYXL1-SRRM3*, and *C1orf141-IL23R*, that conferred susceptibility to sarcoidosis. Our findings suggest that genetic control, through genetic polymorphisms in *CCL24*, *POR*, and *IL23R*, may have an important physiological role in the development and progression of sarcoidosis.

## Methods

**Subjects**. A total of 700 unrelated patients with sarcoidosis and 886 unrelated healthy controls, all of Japanese descent, were enrolled in the GWAS. For replication, we included additional samples from Japan (931 cases and 1,042 controls) and the Czech Republic (265 cases and 264 controls; all Caucasians from Central Europe). Sarcoidosis was diagnosed in Japanese patients according to the

"Diagnostic Criteria and Guidelines for Sarcoidosis" developed by the Japanese Society of Sarcoidosis and Other Granulomatous Disorders (JSSOG 2007)[39]. Patients were recruited at Yokohama City University, JR Sapporo Hospital, Kyoto Health Care Laboratory, Aichi Medical University, JR Tokyo General Hospital, Hokkaido University, Tokyo University, Tokyo Medical and Dental University, Fujita Health University, Nippon Medical School, Kono Medical Clinic, Tokyo Medical University, Hamamatsu University School of Medicine, Japanese Red Cross Medical Center, Keio University, Kumamoto City hospital, National Defense Medical College, Yuasa Eye Clinic, Nishibeppu National Hospital, JCHO Osaka Hospital, Sapporo Medical University, and Health Sciences University of Hokkaido. All Japanese controls were healthy volunteers unrelated to each other or to the patients. Sarcoidosis was diagnosed in Czech patients according to the criteria in the ATS/ERS/WASOG International Consensus Statement[2], and they were recruited at a single tertiary referral center (Palacky University Hospital, Olomouc, Czech Republic). The Czech controls were healthy participants in the bone marrow donor registry, recruited from the same region as the patients. All Czech participants were Caucasians from Central Europe. The absence of lung disease in the controls was confirmed by a health questionnaire and interview, which emphasized family history and symptoms of respiratory disease. Sarcoidosis was staged according to findings on CXRs, as follows: stage 0, normal; stage I, bilateral hilar lymphadenopathy (BHL); stage II, BHL with pulmonary infiltrates; stage III, infiltrates without BHL; stage IV, pulmonary fibrosis. Stages III and IV are the most advanced forms of the disease; they are characterized by parenchymal involvement and fibrosis, with a low rate of spontaneous resolution. This study was approved by the Ethics Committee of Yokohama City University (approval number: A150122003). The study complied with the guidelines of the Declaration of Helsinki. The study details were explained to all patients and controls, and written informed consent was obtained for genetic screening.

We used the QIAamp DNA Blood Maxi Kit (QIAGEN, Hilden, Germany) to collect peripheral blood lymphocytes and extract genomic DNA from peripheral blood cells. Procedures were performed under standardized conditions to prevent variation in DNA quality.

**Genome-wide association study and quality control procedures**. Genotyping was performed with the Illumina HumanOmniExpress chip (727,413 SNPs) using the standard protocol recommended by Illumina (San Diego, CA, USA). SNPs were excluded, based on the following criteria: a call rate <98%; the rates of missing data were significantly different between cases and controls ($P < 1.0 \times 10^{-6}$); an overall minor allele frequency <5%; and a significant deviation from Hardy-Weinberg equilibrium (HWE) in controls ($P < 0.0001$). For the samples, we set the minimum SNP call rate to 99%. In addition, cryptic relatedness between samples was estimated based on identity by descent; closely related samples with a pi-hat >0.1875 were eliminated. We also excluded samples with excessive heterozygosity (>5 standard deviations from the mean sample heterozygosity). Finally, a total of 530,466 autosomal SNPs (685 cases and 847 controls) that passed through the filters were used for subsequent statistical analyses. Quantile–quantile plots and the genomic inflation factor ($\lambda$) were used to assess the presence of systematic bias in the test statistics, due to potential population stratification. We also performed a principal component analysis to visualize the degree of genetic stratification between cases and controls.

**Replication of the GWAS findings**. We tested the lead SNPs from each of the GWAS-identified loci with the Illumina GoldenGate assay according to the standard protocol recommended by Illumina for replication in 1,973 Japanese and 529 Czech individuals. For samples, we set the minimum SNP call rate to 99%, and we obtained genotyping results for 1,949 Japanese and 508 Czech individuals. For a meta-analysis of the GWAS discovery and two replication cohorts, we used SNPs that had a consistent direction and magnitude of allelic effect across all three cohorts.

**Imputation analysis and fine-mapping for candidate loci**. The genotypes of the GWAS set were imputed with the MACH v1.0 program (http://www.sph.umich.edu/csg/abecasis/MACH/index.html)[40,41]. For the reference panel, we used the 1000 Genomes Phase 3 data sets (http://www.1000genomes.org/) for 315 East-Asian samples, which included 104 Japanese individuals in Tokyo (JPT); 103 Han Chinese individuals in Beijing (CHB); and 108 Southern Han Chinese (CHS)[42]. All imputed SNPs were filtered with the quality control parameters (HWE $P > 0.001$, minor allele frequency >0.01, and squared correlation between imputed and true genotypes [Rsq] >0.3).

To highlight the potentially causal SNPs in the identified loci, we performed fine-mapping with FINEMAP v1.2 software[13]. This software uses a shotgun stochastic search algorithm. We ran FINEMAP with the assumption that there would only be one causal variant in each locus. We calculated the posterior inclusion probabilities and $\log_{10}$ Bayes factor to assess the causality of each SNP within the identified loci.

**Imputation analysis for HLA alleles**. The genotypes of 6,181 SNPs located in the *HLA* region were selected from the GWAS data. For imputation, we used SNP2HLA v1.0.3 (https://www.broadinstitute.org/mpg/snp2hla/)[43] and a reference

panel of 530 pan-Asian samples[44]. Imputed *HLA* alleles with $r^2 > 0.8$ were included in the association analysis. We also directly genotyped *HLA-DRB1* alleles for 700 cases with Luminex reverse sequence-specific oligonucleotides and bead kits (One Lambda, Canoga Park, CA, USA). The concordance rate between imputed and direct genotyping was 96.4% for four-digit *HLA-DRB1* types.

**Expression analysis**. Whole blood was collected from 342 healthy Japanese individuals in PAXgene Blood RNA tubes (Becton Dickinson, Heidelberg, Germany). Total RNA was extracted from whole blood with the PAXgene Blood RNA Kit (QIAGEN) according to the manufacturers' protocols. cDNA was synthesized with the SuperScrip IV VILO Master Mix (Thermo Fisher Scientific, Waltham, MA, USA).

Expression analyses for the targeted genes were performed with the TaqMan Gene Expression Assays (Applied Biosystems, Foster City, CA, USA) and the StepOnePlus Real-Time PCR System (Applied Biosystems), according to the manufacturers' protocols. Relative expression was quantified with the comparative Ct (ΔΔCt) method, and *GAPDH* gene expression served as an endogenous reference. Expression data were also extracted from the GTEx Portal online database (version 8, https://www.gtexportal.org/home/)[14] on March 5, 2020 and *P*-value significance threshold was applied as described in the original report[45].

We also analyzed colocalization with eCAVIAR[15] to identify potentially causal SNPs that were responsible for in both GWAS and eQTL signals within the identified loci. For the colocalization analysis, we used each *Z*-score obtained from the genetic association results for the Japanese GWAS discovery data set and from the whole-blood eQTL results for the healthy Japanese data set; we also used each LD structure, based on the genotype data from the Japanese GWAS and eQTL data sets. We selected SNPs with *Z*-scores > 3 and >2, in the genetic association and eQTL analyses, respectively. eCAVIAR estimated the colocalization posterior probability (CLPP), which was the probability that the same variant was causal in both the GWAS and eQTL data. We considered SNPs with CLPP > 0.01 significant, as recommended in the original report[15].

**Statistics and reproducibility**. The quality control procedures for GWAS, all association analyses, the stepwise regression analyses, and the eQTL data analyses were carried out with SNP & Variation Suite software (version 8.8.3, Golden Helix, Bozeman, MT, USA). Association analyses were performed with an additive genetic model in the Japanese GWAS discovery cohort (685 cases and 847 controls), and the obtained *P*-values were adjusted for the first five ancestry principal components and genomic inflation ($P_{GC}$) in the GWAS discovery stage to correct for the potential presence of population stratification. In the replication analysis using the Japanese (907 cases and 1042 controls) and Czech cohorts (252 cases and 256 controls), we also performed association analyses under an additive genetic model. Meta-analyses for the various populations were performed with the inverse variance-weighted fixed-effect model with the META v1.7 program[46]. A Cochran's *Q* test was used to assess heterogeneity across populations. The genome-wide significance threshold was set to $P_{GC} < 5 \times 10^{-8}$. Regional association plots for the target regions were generated with LocusZoom v1.3[47]. The LD structures of the targeted regions were inferred with LocusZoom v1.3 and Haploview 4.1 software[48].

**Reporting summary**. Further information on research design is available in the Nature Research Reporting Summary linked to this article.

## Data availability
The majority of the data generated or analyzed during this study are included in this published article and its Supplementary Information files. The summary statistics of the GWAS are included in Supplementary Data 3. Other data supporting the findings of this study are available from the corresponding author on reasonable request.

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

## Acknowledgements

We sincerely thank the patients with sarcoidosis that participated in this study. We also thank Lisa Endo and Reiko Nanba for their excellent technical assistance. This work was supported by the Japanese Society of Sarcoidosis and Other Granulomatous Disorders; the Study Group on Diffuse Pulmonary Disorders from the Ministry of Health, Labour, and Welfare, Japan; JSPS KAKENHI Grants 24790338, 26860226, and 25861643; grants from the Japan Foundation for Applied Enzymology; and grants from the Czech government (NV18-05-00134) and Palacky University (IGA_PU_LF 2018_015, 2019_009).

## Author contributions

Study design: A.M., M. Ishihara, M.P., and N.M. Analysis: A.M., M. Ishihara, M.P., K. Yamamoto, Masaki Takeuchi, F.M., T. Yamane., E.S., A.Y., A.I., M.O., S.B., E.F.R., D.L.K., and N.M. Sample procurement and data generation: A.M., M. Ishihara, M.P., K. Yamamoto, Masaki Takeuchi, F.M., V.K., A.B., T. Yamane., E.S., A.Y., A.I., M.O., K. Yatsu, N.S., S.N., E.Y., T. Yamaguchi, K.N., T. Kaburaki, H.T., S.M., J.H., K.K., H.G., T.S., S.-i.M., Y. Ando, S.T., Masaru Takeuchi, T. Yuasa, K.S., N.O., M.H., N.K., Y.S., N.H., Y. Asukata, T. Kawagoe, I.K., M. Ishido, H.I., M.M., S.O., S.B., E.F.R., D.L.K., and N.M. Writing: A.M., M. Ishihara, M.P., S.B., E.F.R., D.L.K., and N.M. Finalization of the manuscript: A.M., M.P., and N.M.

## Competing interests

The authors declare no competing interests.

## Additional information

Akira Meguro[1], Mami Ishihara[1], Martin Petrek[2], Ken Yamamoto[3,4], Masaki Takeuchi[1,5], Frantisek Mrazek[6], Vitezslav Kolek[7], Alzbeta Benicka[2], Takahiro Yamane[1], Etsuko Shibuya[1], Atsushi Yoshino[1], Akiko Isomoto[4], Masao Ota[1,8,9,10], Keisuke Yatsu[11], Noriharu Shijubo[12], Sonoko Nagai[13], Etsuro Yamaguchi[14], Tetsuo Yamaguchi[15], Kenichi Namba[16], Toshikatsu Kaburaki[17], Hiroshi Takase[18], Shin-ichiro Morimoto[19], Junko Hori[20], Keiko Kono[21], Hiroshi Goto[22], Takafumi Suda[23], Soichiro Ikushima[24], Yasutaka Ando[25,26], Shinobu Takenaka[27], Masaru Takeuchi[28], Takenosuke Yuasa[29], Katsunori Sugisaki[30], Nobuyuki Ohguro[31], Miki Hiraoka[32], Nobuyoshi Kitaichi[16,33], Yukihiko Sugiyama[34], Nobuyuki Horita[5,35], Yuri Asukata[1], Tatsukata Kawagoe[1], Ikuko Kimura[1], Mizuho Ishido[1], Hidetoshi Inoko[9,10,36], Manabu Mochizuki[18], Shigeaki Ohno[16,33], Seiamak Bahram[9,10,37], Elaine F. Remmers[5], Daniel L. Kastner[5] & Nobuhisa Mizuki[1 ✉]

[1]Department of Ophthalmology and Visual Science, Yokohama City University Graduate School of Medicine, 3-9 Fukuura, Kanazawa-ku, Yokohama, Kanagawa 236-0004, Japan. [2]Department of Pathological Physiology, Faculty of Medicine and Dentistry, Palacky University, Hnevotinska Str., 77515 Olomouc, Czech Republic. [3]Department of Medical Biochemistry, Kurume University School of Medicine, 67 Asahimachi, Kurume, Fukuoka 830-0011, Japan. [4]Division of Genome Analysis, Institute of Bioregulation, Kyushu University, 3-1-1 Maidashi, Higashi-ku, Fukuoka, Fukuoka 812-8582, Japan. [5]Inflammatory Disease Section, National Human Genome Research Institute, National Institutes of Health, 10 Center Drive, 10 CRC East/B2-5235, Bethesda, MD 20892-1849, USA. [6]Department of Immunology, Faculty of Medicine and Dentistry, Palacky University, I.P.Pavlova Str. 6, 77520 Olomouc, Czech Republic. [7]Department of Respiratory Medicine, Faculty of Medicine and Dentistry, Palacky University, I. P. Pavlova Str. 6, 77900 Olomouc, Czech Republic. [8]Division of Hepatology and Gastroenterology, Department of Medicine, Shinshu University School of Medicine, 3-1-1 Asahi, Matsumoto, Nagano 390-8621, Japan. [9]INSERM Franco-Japanese "Laboratoire International Associé"

(LIA) Nextgen HLA Laboratory, Strasbourg, France. [10]INSERM Franco-Japanese "Laboratoire International Associé" (LIA) Nextgen HLA Laboratory, Nagano, Japan. [11]Department of Medical Science and Cardiorenal Medicine, Yokohama City University School of Medicine, 3-9 Fukuura, Kanazawa-ku, Yokohama, Kanagawa 236-0004, Japan. [12]Department of Respiratory Medicine, Japan Railway Sapporo Hospital, Higashi-1, Kita-3, Chuo-ku, Sapporo 060-0033, Japan. [13]Kyoto Central Clinic/Clinical Research Center, 56-58 Masuyacho Sanjo-Takakura, Nakagyo-ku, Kyoto 604-8111, Japan. [14]Division of Respiratory Medicine and Allergology, Aichi Medical University, 21 Karimata, Yazako, Nagakute-cho, Aichi-gun, Aichi 480-1195, Japan. [15]Department of Respiratory Medicine, Japan Railway Tokyo General Hospital, 2-1-3 Yoyogi, Shibuya-ku, Tokyo 151-0053, Japan. [16]Department of Ophthalmology, Faculty of Medicine and Graduate School of Medicine, Hokkaido University, N15, W7, Kita-ku, Sapporo, Hokkaido 060-8638, Japan. [17]Department of Ophthalmology, University of Tokyo School of Medicine, 7-3-1, Hongo, Bunkyo-ku, Tokyo 113-8655, Japan. [18]Department of Ophthalmology and Visual Science, Tokyo Medical and Dental University Graduate School of Medicine, 1-5-45 Yushima, Bunkyo-ku, Tokyo 113-8519, Japan. [19]Division of Cardiology, Department of Internal Medicine, Fujita Health University School of Medicine, 1-98 Dengakugakubo, Kutsukakecho, Toyoake, Aichi 470-1192, Japan. [20]Department of Ophthalmology, Nippon Medical School, 1-1-5 Sendagi, Bunkyo-ku, Tokyo 113-8602, Japan. [21]Department of Ophthalmology, Kono Medical Clinic, 3-30-28 Soshigaya, Setagaya-ku, Tokyo 157-0072, Japan. [22]Department of Ophthalmology, Tokyo Medical University, 6-7-1 Nishishinjuku, Shinjuku-ku, Tokyo 160-0023, Japan. [23]Second Division, Department of Internal Medicine, Hamamatsu University School of Medicine, 1-20-1 Handayama, Hamamatsu, Shizuoka 431-3192, Japan. [24]Department of Respiratory Medicine, Japanese Red Cross Medical Centre, 4-1-22 Hiroo, Shibuya-ku, Tokyo 150-8953, Japan. [25]Department of Ophthalmology, Kitasato Institute Hospital, 5-9-1 Shirokane, Minato-ku, Tokyo 108-8642, Japan. [26]Department of Ophthalmology, Keio University School of Medicine, 35 Shinanomachi, Shinjuku-ku, Tokyo 160-0016, Japan. [27]Department of Respiratory Diseases, Kumamoto City Hospital, 1-1-60 Kotoh, Kumamoto, Kumamoto 862-8505, Japan. [28]Department of Ophthalmology, National Defense Medical College, 3-2 Namiki, Tokorozawa, Saitama 359-8513, Japan. [29]Yuasa Eye Clinic, 3-1-1 Nishimoto-cho, Nishi-ku, Osaka 550-0005, Japan. [30]Department of Internal Medicine, National Hospital Organization Nishibeppu National Hospital, 4548 Oaza-Tsurumi, Beppu, Oita 874-0840, Japan. [31]Department of Ophthalmology, Japan Community Health care Organization Osaka Hospital, 4-2-78 Fukushima, Fukushima-ku, Osaka 553-0003, Japan. [32]Department of Ophthalmology, School of Medicine, Sapporo Medical University, S1 W16 Chuo-ku, Sapporo, Hokkaido 060-8543, Japan. [33]Department of Ophthalmology, Health Sciences University of Hokkaido, Ainosato 2-5, Kita-ku, Sapporo, Hokkaido 002-8072, Japan. [34]Division of Pulmonary Medicine, Department of Medicine, Jichi Medical University, 3311-1 Yakushiji, Shimotsuke, Tochigi 329-0498, Japan. [35]Department of Pulmonology, Yokohama City University Graduate School of Medicine, 3-9 Fukuura, Kanazawa-ku, Yokohama, Kanagawa 236-0004, Japan. [36]Department of Molecular Life Science, Division of Molecular Medical Science and Molecular Medicine, Tokai University School of Medicine, 143 Shimokasuya, Isehara, Kanagawa 259-1193, Japan. [37]Plateforme GENOMAX, Laboratoire d'ImmunoRhumatologie Moléculaire, INSERM UMR_S1109, LabEx Transplantex, Centre de Recherche d'Immunologie et d'Hématologie. Faculté de Médecine, Fédération Hospitalo-Universitaire (FHU) OMICARE, Fédération de Médecine Translationnelle de Strasbourg (FMTS), Université de Strasbourg, Strasbourg, France. ✉email: mizunobu@yokohama-cu.ac.jp

