## [Peer Review File · Communications Biology]

Reviewers' comments:

Reviewer #1 (Remarks to the Author):

This manuscript by Meguro et al. reports a trans-ethnic GWAS of sarcoidosis across Japanese and Czech populations. The authors discovered three genome-wide significant non-HLA loci underlying this disease, of which two are novel findings. They conducted eQTL analyses to predict candidate causal genes underlying the identified loci. The manuscript is very interesting and furthers our understanding of the genetic basis of sarcoidosis. I believe it warrants publication in the journal if the following issues can be addressed.

Major:

Lines 239-240, p. 14: The authors claim that "the disease-associated SNPs were located in non-coding regions." I am not objecting to this, but this was not supported by fine-mapping analysis. The observation that the most strongly associated SNPs land in the non-coding regions is not sufficient here. I recommend the authors to run a credible SNP set analysis.

Lines 242-248, p. 14 (eQTL analysis, Supplementary Table 5): The association of SNPs with expression levels does not prove the regulatory role of these SNPs. It is well known that the causal SNPs for disease risk and eQTL can be distinct from each other, and the observed eQTL association can be sometimes driven by LD between two distinct causal SNPs. Colocalization analysis (e.g. coloc, eCAVIAR or JLIM) has to be done here to show that eQTL and disease risk are driven by the shared genetic effect.

Supplementary Table 5. The authors examined eQTLs in whole blood samples of the study cohort and GTEx tissues. On the GTEx, they report eQTLs only in transformed fibroblasts and esophagus. Did the authors observe the eQTLs in GTEx whole blood? How about naive T cells (e.g. BLUEPRINT)? Are the cell and tissue types examined here relevant for the proposed disease pathway? The manuscript will benefit from further discussion and clarification of this issue.

Minor:

Line 167, p. 10: I would recommend avoiding the misleading expression "most significant association" here since it is not significant.

Lines 170-175, p.10 and Supplementary Table 3: It seems that all SNPs in previously identified risk loci were tested and the minimum p-value in each locus was examined to test for replication of known hits. The lead SNP of original studies (or their close proxy) has to be directly tested, otherwise, the degree of multiple testing in each locus has to be accounted for.

Line 254, p. 15: Is this trend significant?

Reviewer #2 (Remarks to the Author):

Dr Meguro and co-authors present a genome-wide association study using the Illumina OmniExpress beadarray to study the genetic architecture underlying sarcoidosis. From a starting number of 700 cases and 886 controls from Japan, QC procedures were applied, resulting in 685 cases and 847

controls for downstream association analysis. SNP associations surpassing allelic $P < 0.01$ were followed up in additional case-control series from Japan and the Czech Republic. The authors confirmed association at the broad HLA locus, and identified 3 non-HLA gene-loci. These findings provide considerable additional biological insights into the disease process underlying sarcoidosis.

It was a pleasure reading this manuscript. In particular, I appreciated very much the robust discussion at the IL23R locus with reference to other auto-immune phenotypes, showing that the association with sarcoid was distinct, and that the sarcoid association could actually be shared with VKH!. Also, the authors took great effort in confirming the previous association at ANXA11, which was first described in European ancestry populations. The very detailed dissection at the HLA locus (separating protective and risk haplotypes, and only scoring the genome-wide significant ones) is to be highly commended.

I have some minor technical suggestions for the authors to help improve the accessibility of the manuscript to the readers of Communications Biology.

1. A principal component plot to visualize the degree of genetic stratification between patients with sarcoidosis and unaffected individuals is suggested. This is in view of the rather high genomic inflation factor (1.05) in the context of a relatively modest sample size of 685 cases and 847 controls. In addition, SNP rs7871278 on chr 9p21 which almost reached genome-wide significance in the GWAS stage ($P < 5.8 \times 10^{-8}$) failed to replicate in the follow up stage. Usually, SNPs with $P < 1 \times 10^{-7}$ have a >90% chance of being true positives if vigorous QC and control of stratification have been applied. This could be a point of discussion for the authors to consider.

2. The application of the allele-based test (is it the allelic chi-squared test?) to measure the association between SNP genotypes and case-control status could have resulted in some inflation of P-values. I humbly suggest using logistic regression with correction for the top ancestry principal components.

3. Page 24, line 423. a frequency cut off of 5% was used for genotyped SNPs. Why is this threshold different from the threshold applied for imputed SNPs (page 25, line 446; 1%)?

4. In the same vein, the sample filtering threshold appeared to be different in the GWAS discovery stage (SNP call rate at 99%, page 24, line 424) compared to the replication stage (page 24, line 435, 95%).

5. As a quality control procedure, were samples with excessive heterozygosity removed? If so, at what threshold were they removed?

6. For meta-analysis, can the authors include the Breslow-Day estimate of heterogeneity? (or if they use logistic regression and inverse variance meta-analysis, the I² index of heterogeneity)

CC Khor

Reviewers' comments:

Reviewer #1 (Remarks to the Author):

This manuscript by Meguro et al. reports a trans-ethnic GWAS of sarcoidosis across Japanese and Czech populations. The authors discovered three genome-wide significant non-HLA loci underlying this disease, of which two are novel findings. They conducted eQTL analyses to predict candidate causal genes underlying the identified loci. The manuscript is very interesting and furthers our understanding of the genetic basis of sarcoidosis. I believe it warrants publication in the journal if the following issues can be addressed.

Major:

Lines 239-240, p. 14: The authors claim that "the disease-associated SNPs were located in non-coding regions." I am not objecting to this, but this was not supported by fine-mapping analysis. The observation that the most strongly associated SNPs land in the non-coding regions is not sufficient here. I recommend the authors to run a credible SNP set analysis.

Response: As recommended, we performed a fine-mapping analysis with FINEMAP software. We added the results to the revised manuscript, in the 7th paragraph of the Results section, Figure 2, and Supplementary Table 5.

Lines 242-248, p. 14 (eQTL analysis, Supplementary Table 5): The association of SNPs with expression levels does not prove the regulatory role of these SNPs. It is well known that the causal SNPs for disease risk and eQTL can be distinct from each other, and the observed eQTL association can be sometimes driven by LD between two distinct causal SNPs. Colocalization analysis (e.g. coloc, eCAVIAR or JLIM) has to be done here to show that eQTL and disease risk are driven by the shared genetic effect.

Response: As suggested, we performed a colocalization analysis. The results are shown in the 9th paragraph of the Results section and Supplementary Table 7.

Supplementary Table 5. The authors examined eQTLs in whole blood samples of the study cohort and GTEx tissues. On the GTEx, they report eQTLs only in transformed fibroblasts and esophagus. Did the authors observe the eQTLs in GTEx whole blood? How about naive T cells (e.g. BLUEPRINT)? Are the cell and tissue types examined here are relevant for the proposed disease pathway? The manuscript will benefit from further discussion and clarification of this issue.

Response: We have updated the eQTL results obtained with the latest GTEx Portal (version 8) in the revised manuscript. We show the results for CCL24 and POR in all tissues tested in the GTEx Portal in Supplementary Figures 5 and 6. We commented on the eQTL findings in the

Results and Discussion sections.

Minor:

Line 167, p. 10: I would recommend avoiding the misleading expression "most significant association" here since it is not significant.

Response: This has been corrected.

Lines 170-175, p.10 and Supplementary Table 3: It seems that all SNPs in previously identified risk loci were tested and the minimum p-value in each locus was examined to test for replication of known hits. The lead SNP of original studies (or their close proxy) has to be directly tested, otherwise, the degree of multiple testing in each locus has to be accounted for.

Response: As suggested, we have applied the Bonferroni correction for multiple testing in previously identified risk loci. The corrected results were added to the revised manuscript.

Line 254, p. 15: Is this trend significant?

Response: For CCL24 rs4728493, the trend was significant (Mantel-extension test: $P=0.045$). On the other hand, for STYXL1-SRRM3 rs112463197, the trend was not significant. We have stated these results in the Results section.

Note: We used LocusZoom software to show the association plots and LDs in the targeted regions in the revised manuscript (Figure 2 and Supplementary Figure 3).

Reviewer #2 (Remarks to the Author):

Dr Meguro and co-authors present a genome-wide association study using the Illumina OmniExpress beadarray to study the genetic architecture underlying sarcoidosis. From a starting number of 700 cases and 886 controls from Japan, QC procedures were applied, resulting in 685 cases and 847 controls for downstream association analysis. SNP associations surpassing allelic $P < 0.01$ were followed up in additional case-control series from Japan and the Czech Republic. The authors confirmed association at the broad HLA locus, and identified 3 non-HLA gene-loci. These findings provide considerable additional biological insights into the disease process underlying sarcoidosis.

It was a pleasure reading this manuscript. In particular, I appreciated very much the robust discussion at the IL23R locus with reference to other auto-immune phenotypes, showing that the association with sarcoid was distinct, and that the sarcoid association could actually be shared with VKH!. Also, the

authors took great effort in confirming the previous association at ANXA11, which was first described in European ancestry populations. The very detailed dissection at the HLA locus (separating protective and risk haplotypes, and only scoring the genome-wide significant ones) is to be highly commended.

I have some minor technical suggestions for the authors to help improve the accessibility of the manuscript to the readers of Communications Biology.

1. A principal component plot to visualize the degree of genetic stratification between patients with sarcoidosis and unaffected individuals is suggested. This is in view of the rather high genomic inflation factor (1.05) in the context of a relatively modest sample size of 685 cases and 847 controls. In addition, SNP rs7871278 on chr 9p21 which almost reached genome-wide significance in the GWAS stage ($P < 5.8 \times 10^{-8}$) failed to replicate in the follow up stage. Usually, SNPs with $P < 1 \times 10^{-7}$ have a >90% chance of being true positives if vigorous QC and control of stratification have been applied. This could be a point of discussion for the authors to consider.

Response: We performed a principal component analysis and found that the cases and controls in the GWAS discovery stage were genetically well matched. The principal component plot has been added to Supplementary Figure 2.

We calculated P-values adjusted for genomic inflation in the GWAS discovery stage according to Comment 2 from Reviewer 2. After adjusting the P-values, there was no SNP with $P_{GC} < 1 \times 10^{-7}$ in the discovery stage outside the HLA region (the strongest signal was rs7871278: $P_{GC} = 1.3 \times 10^{-7}$). Therefore, we did not add a discussion about SNPs with $P < 1 \times 10^{-7}$ in the GWAS stage. Instead, we show the results (P-value and odds ratio) of rs7871278 in the replication stage (last sentence of the fourth paragraph of the Results section).

2. The application of the allele-based test (is it the allelic chi-squared test?) to measure the association between SNP genotypes and case-control status could have resulted in some inflation of P-values. I humbly suggest using logistic regression with correction for the top ancestry principal components.

Response: We generated P-values adjusted for genomic inflation in the GWAS discovery stage. The adjustment of P-values did not significantly affect our findings in the current study. We have added the adjusted P-values (P_{GC}) to the revised manuscript.

3. Page 24, line 423. a frequency cut off of 5% was used for genotyped SNPs. Why is this threshold different from the threshold applied for imputed SNPs (page 25, line 446; 1%)?

Response: We used a minor allele frequency cut-off of 5% for genotyped SNPs in the GWAS discovery stage with 700 cases and 886 controls, because calling low/rare genotypes (such as those with less than 5% frequencies) in a small or medium sample size is subject to frequent errors. On the other hand, we applied the cut-off of 1% to the imputed SNPs, because

imputation was performed only for the identified three loci, and we could assess the imputation quality of the imputed SNPs.

4. In the same vein, the sample filtering threshold appeared to be different in the GWAS discovery stage (SNP call rate at 99%, page 24, line 424) compared to the replication stage (page 24, line 435, 95%).

Response: The description on page 24, line 435 was not appropriate. In the replication stage, with the GoldenGate assay, we excluded samples with SNP call rates less than 95%. We performed the assay three times; however, the SNP call rate did not reach 95% in these excluded samples, due to the low quality of the DNA samples. On the other hand, the samples included in the replication stage had a high SNP call rate, more than 99%. In other words, “For samples, we set the minimum SNP call rate to 99%” in the replication stage. We have updated this description in the revised manuscript.

5. As a quality control procedure, were samples with excessive heterozygosity removed? If so, at what threshold were they removed?

Response: Thank you for this comment. We did remove samples with excessive heterozygosity (> 5 standard deviations from the mean heterozygosity). We have added a statement in the quality control procedure to indicate that we excluded samples with excessive heterozygosity (Methods section).

6. For meta-analysis, can the authors include the Breslow-Day estimate of heterogeneity? (or if they use logistic regression and inverse variance meta-analysis, the I² index of heterogeneity)

Response: We performed a Cochran’s Q test to assess heterogeneity across populations. We have included those results in Table 1 and Supplementary Tables 7 and 8 of the revised manuscript.

Note: We used LocusZoom software to show the association plots and LDs in the targeted regions in the revised manuscript (Figure 2 and Supplementary Figure 3).

Reviewers' comments:

Reviewer #1 (Remarks to the Author):

1. I share the other review's concern on the issue of using an allele-based test without covariate correction. The global adjustment for genomic inflation is helpful but does not entirely address this issue. Population stratification might not uniformly confound p-values of association across the genome to the same degree. Thus, stochastically, p-values of some loci might become more severely inflated than the rest. I strongly recommend the authors to double-check the analysis by applying logistic regression with ancestry PCs as well as other potential confounders as covariates.

2. I am puzzled by the author's eCAVIAR colocalization results. When GWAS lead SNPs are in tight LD with colocalizing SNPs ($r^2 \geq 0.95$ in Japanese), why are the CLIP posteriors so low for the lead SNPs compared to other SNPs (Supp. Table 7)? This doesn't make sense to me. Is this because the r^2 between these SNPs is low in Czech and meta-analyzed GWAS statistics are used for eCAVIAR? This is okay, but I wonder how the reference LD was set for the colocalization analysis. There is no description of the detailed parameter settings in the Methods section. eCAVIAR has to be run with the LD matrices matching the GWAS and eQTL populations, respectively.

3. It is also not clear whether the colocalization analysis was done with multi-tissue eQTL or single-tissue eQTL data. If it's done with the former, this is not accurate because different tissues may have different cis-regulatory variation driving the eQTL effects. I would recommend testing for the colocalization with single-tissue eQTL data of most relevant tissue.

The authors addressed my other previous concerns.

Reviewer #2 (Remarks to the Author):

Dr Meguro and co-authors present a very thorough and transparent response to the first round of review.

I have no additional concerns.

CC Khor

Reviewers' comments:

Reviewer #1 (Remarks to the Author):

1. I share the other review's concern on the issue of using an allele-based test without covariate correction. The global adjustment for genomic inflation is helpful but does not entirely address this issue. Population stratification might not uniformly confound p-values of association across the genome to the same degree. Thus, stochastically, p-values of some loci might become more severely inflated than the rest. I strongly recommend the authors to double-check the analysis by applying logistic regression with ancestry PCs as well as other potential confounders as covariates.

Response 1: Thank you for this comment. We have now performed association analyses with an additive model, and we adjusted the P-values for both the ancestry principal components and genomic inflation, to correct for the possible presence of population stratification in the GWAS discovery stage. Adjusting the P-values did not significantly affect our findings in the current study. We have now shown the P-values adjusted for both the ancestry PCs and genomic inflation in the revised manuscript.

2. I am puzzled by the author's eCAVIAR colocalization results. When GWAS lead SNPs are in tight LD with colocalizing SNPs ($r^2 \geq 0.95$ in Japanese), why are the CLIP posteriors so low for the lead SNPs compared to other SNPs (Supp. Table 7)? This doesn't make sense to me. Is this because the r^2 between these SNPs is low in Czech and meta-analyzed GWAS statistics are used for eCAVIAR? This is okay, but I wonder how the reference LD was set for the colocalization analysis. There is no description of the detailed parameter settings in the Methods section. eCAVIAR has to be run with the LD matrices matching the GWAS and eQTL populations, respectively.

Response 2: We really appreciate your insight. In the original colocalization analysis, the ethnicity of the population used in the GWAS statistics did not match the ethnicity of the population used in the eQTL statistics; i.e., we used the GWAS statistics for the Japanese discovery cohort and the eQTL statistics for Europeans obtained from the GTEx Portal database. To address this limitation, we matched the ethnicity between the populations used for the GWAS and eQTL statistics by using the GWAS statistics for the Japanese discovery cohort and the whole-blood eQTL statistics for the healthy Japanese dataset. In the revised colocalization analysis, the GWAS lead SNPs in CCL24, STYXL1-SRRM3, and IL23R showed significant colocalization (CLPP>0.01). We have now shown the details of the colocalization analysis in the Methods section and updated the results in the revised manuscript.

3. It is also not clear whether the colocalization analysis was done with multi-tissue eQTL or single-tissue eQTL data. If it's done with the former, this is not accurate because different tissues may have different cis-regulatory variation driving the eQTL effects. I would recommend testing for the

colocalization with single-tissue eQTL data of most relevant tissue.

Response 3: We only have access to RNA extracted from whole-blood samples, and not from other tissues/organs. Therefore, we used the single-tissue eQTL data from the whole-blood samples. Sarcoidosis is a systemic inflammatory disease that affects multiple organs in the body, including the lungs, skin, eyes, lymph nodes, liver, spleen, central and peripheral nervous systems, heart, salivary glands, and muscles. Because the whole-blood transcriptional profile in sarcoidosis reportedly reflects transcriptional abnormalities observed in the specific diseased organs, we believe that our colocalization findings from the whole-blood samples reflected the expression profiles of the specific diseased organs.

On the other hand, the GTEx Portal database has eQTL data from many tissues in Europeans. However, the sample size of our Czech cohort for the genetic association analysis was insufficient (low statistical power) to perform a colocalization analysis. Therefore, for the revised manuscript, we did not perform the analysis with the European eQTL data from the GTEx Portal and the genetic association data from the Czech samples.

Reviewer #2 (Remarks to the Author):

Dr Meguro and co-authors present a very thorough and transparent response to the first round of review.

I have no additional concerns.

REVIEWERS' COMMENTS:

Reviewer #1 (Remarks to the Author):

The authors addressed the rest of my previous concerns.